# THE IMPORTANCE AND EFFECTIVENESS OF LEARNING THROUGH SOCIAL FEEDBACK

**Natasha Jaques**[12]**, Jesse Engel**[2]**, David Ha**[2]**, Fred Bertsch**[2]**, Rosalind Picard**[1]**, Douglas Eck**[2]
[1]Massachusetts Institute of Technology, Cambridge, MA 02139, USA
[2]Google Brain, Mountain View, CA 94043, USA
`jaquesn@mit.edu`, `jesseengel@google.com`, `hadavid@google.com`,
`fredbertsch@google.com`, `picard@mit.edu`, `deck@google.com`

## ABSTRACT

In the quest towards general artificial intelligence (AI), researchers have explored developing loss functions that act as intrinsic motivators in the absence of external rewards. This paper argues that such research has overlooked an important and useful intrinsic motivator: social interaction. We posit that making an AI agent aware of implicit social feedback from humans can allow for faster learning of more generalizable and useful representations, and could potentially impact AI safety. We collect social feedback in the form of facial expression reactions to samples from Sketch RNN, an LSTM-based variational autoencoder (VAE) designed to produce sketch drawings. We use a Latent Constraints GAN (LC-GAN) to learn from the facial feedback of a small group of viewers, and then show in an independent evaluation with 76 users that this model produced sketches that lead to significantly more positive facial expressions. Thus, we establish that implicit social feedback can improve the output of a deep learning model.

## 1 INTRODUCTION

Despite the recent rapid and compelling progress in machine learning and deep learning, modern AI systems are still remarkably far from approximating the intelligence of even simple animals. A notable deficit of current techniques is the degree of explicit supervision required in order to learn, either through labeled samples or well-defined external rewards such as points in a game. The limited scope of such supervision will not enable the development of a generally intelligent AI.

For this reason, some researchers have focused on intrinsic motivators, inherent drives that cause the agent to learn representations that are useful across a variety of tasks and environments. Examples include curiosity (a drive for novelty) (Pathak et al., 2017), and empowerment (a drive for the ability to manipulate the environment) (Capdepuy et al., 2007). However, so far this research has overlooked an important intrinsic motivator for humans: the drive for positive social interactions.

This paper takes the position that making an AI agent intrinsically motivated to obtain a positive social reaction from humans in its environment is an important new research direction. Specifically, the agent should be able to recognize implicit feedback from humans in the form of facial expressions, body language, or tone in voice and text, and optimize for actions that appear to please humans as measured through these signals. Thus, unlike previous approaches to learning from human preferences (e.g. Christiano et al. (2017); Knox & Stone (2009)), explicit supervision from humans is not required for the model to learn. Rather, feedback can be obtained ubiquitously and implicitly, with no additional human effort, through awareness of the non-verbal reactions people naturally provide.

There is substantial evidence that emotion recognition plays an influential role in cognitive development in humans (Kujawa et al., 2014). According to Social Learning Theory (Bandura & Walters, 1977), observing the attitudes and behaviors of others is a central component of how humans learn both intelligent behavior and how to adapt to new situations. It has been argued that social learning is responsible for the rapid cultural evolution of the human species (van Schaik & Burkart, 2011). Given the importance of cultural evolution to humans' technological success, endowing a deep learning agent with the ability to perceive and benefit from this socially exchanged cultural knowledge could allow it to rapidly develop more generalizable knowledge representations.

Crucially, the representations learned by such an agent are more likely to capture dimensions of the task that are relevant to human satisfaction. This has meaningful implications for questions of AI safety; an AI agent motivated by satisfaction expressed by humans will be less likely to take actions against human interest. Such an agent will also be better suited to perform tasks which already involve AI. Imagine if a home assistant could sense when a user responds with an angry or frustrated tone and this acted as a negative incentive, training the algorithm not to repeat the action that led to the user's frustration? Rather than requiring the user to manually train the device, it could learn quickly through passive sensing of the user's emotional state, leading to a more immediately satisfying experience for the user. Finally, some machine learning problems — including the one under investigation in this paper — cannot be solved without human feedback; when the objective function is human aesthetic preference, it cannot be approximated without human input.

In this work we demonstrate the utility of learning through implicit social feedback via an experiment in which samples generated by a deep learning model are presented to people, and their facial expression response is detected. The model is Sketch RNN (Ha & Eck, 2017), an LSTM-based VAE with a Mixture Density Network output, designed to produce sketch drawings. Using a newly developed technique known as Latent Constraints (Engel et al., 2017), we train a Generative Adversarial Network (GAN) to produce VAE embedding vectors $z$ that, when decoded by Sketch RNN, are more likely to produce drawings that lead to positive facial expressions such as smiling. In a rigorous, double-blind evaluation, we show that samples from the social feedback model generate statistically significantly better affective responses than the prior. Thus, this experiment is a first step in demonstrating that deep learning models are able to improve in quality as a result of learning from implicit social feedback.

## 2 METHODS

To gather social feedback, we focused on facial expression recognition, since this is currently one of the most reliable and accurate ways to detect social signals (Senechal et al., 2015). The facial expression detector employed for this project is based on a typical convolutional object-detection network that was re-trained to detect common facial expressions such as amusement and sadness.

To test the hypothesis that facial feedback can improve the outputs of a deep learning model, we sought a model for which the outputs were likely to generate a natural facial expression response. We chose Sketch RNN (Ha & Eck, 2017), a model which generates sequences of strokes that form a sketched image of a common object, vehicle, or animal (see Figure 1). Such sketches were determined to elicit facial responses in initial tests. Sketch RNN is a VAE and MDN which comprises: a) a bidirectional LSTM encoder that projects each input sketch into a latent embedding vector $z$, b) an LSTM decoder which takes $z$ as input and generates a sequence of parameters for c) a Gaussian Mixture Model that generates the $(x, y)$ coordinates of the tip of the pen during each stroke.

Due to the variational constraint, it is straightforward to sample a latent vector $z \sim \mathcal{N}(0, I)$ and feed this into the Sketch RNN decoder to produce a recognizable sketch. This feature allowed us to apply a newly developed technique known as Latent Constraints (Engel et al., 2017) in order to learn to produce sketches likely to lead to positive facial expressions. The latent constraints GAN (LC-GAN) is essentially a GAN applied to the latent embedding space of a VAE. A discriminator $D(z) \rightarrow v$ is trained to estimate the value $v$ of different regions of the latent space; for example, which regions decode to sketches that produced the highest intensity of smiles. A generator $G(z) \rightarrow z'$ is then trained to convert a randomly sampled $z$ into a modified $z'$ that produces a higher $v$. In fact, the generator uses a gating mechanism to control how heavily the original $z$ is modified. The generator loss is $\mathcal{L}_G = -\log D(z')$. Because the latent space of a VAE is already a compressed representation with low covariance among the dimensions, it is straightforward to train a discriminator to learn a value function on $z$, making the LC-GAN well-suited to learn from small sample sizes.

## 3 EXPERIMENT AND RESULTS

To obtain facial feedback at scale, we built a web app that serves samples from Sketch RNN while recording the user's facial expressions with a webcam. The webcam images were fed into the facial-expression detection network and used to compute the intensities of common emotions, including amusement, contentment, surprise, sadness, and concentration. Due to the degree of inter-individual

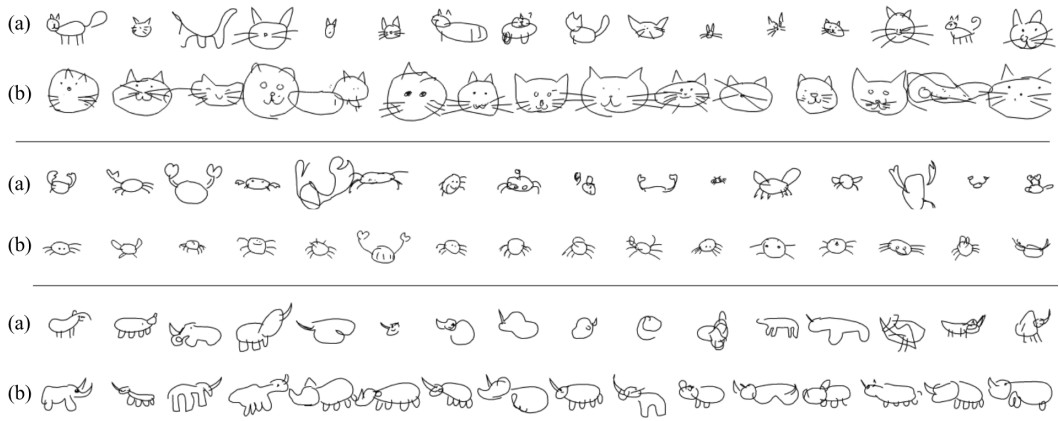

Figure 1: Samples drawn randomly from the cat, crab, and rhinoceros sketch classes, produced by (a) the original Sketch RNN, and (b) the LC-GAN trained on a small amount of social feedback.

variation in users' resting facial expression, these intensities were normalized against each user's average expression to produce value vectors $v$.

Optimizing for user preference requires knowing which facial expressions indicate that the user likes a sketch. Therefore, we ran a preliminary experiment with 7 users and 30 sketches, in which we recorded both facial expressions and users' subjective impressions of each sketch using a 5-point Likert scale. Users' ratings of sketch quality were significantly related to contentment and amusement (smiling), and significantly inversely related to sadness and concentration (frowning), as shown in Table 1. Notably, these

| Emotion metric | $r$ | $p$ |
|---|---|---|
| Contentment | .582 | .001 |
| Amusement | .546 | .002 |
| Concentration | -.576 | .001 |
| Sadness | -.405 | .026 |

Table 1: Correlations between facial expressions and self-reported ratings of sketch quality (all significant at $p = .05$).

results indicate that implicit facial feedback carries an informative signal about the user's preferences. Given the direction of the relationships, we trained the LC-GAN to maximize amusement and contentment, and minimize concentration and sadness.

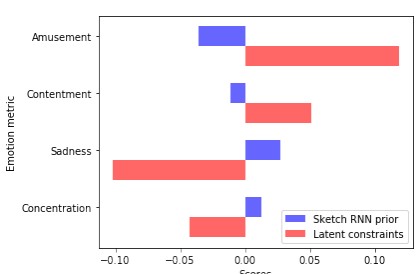

Figure 2: Sketches from the LC-GAN generate more positive expressions.

In the second phase of the study, we used the web app to collect 334 $(z, v)$ pairs from 28 users to train the LC-GAN. Although relatively little data was collected (63-69 samples per sketch class), the LC-GAN was able to effectively optimize for more pleasing sketches. Figure 1 shows the difference between samples produced with the Sketch RNN prior and the LC-GAN. The LC-GAN appears to have learned that people smile more and frown less for cats with larger, smiling faces with whiskers. Similarly, the quality of crab and rhinoceros sketches generated by the LC-GAN appears to be consistently higher.

We objectively evaluated the LC-GAN sketches in a double-blind experiment in which we randomly generated hundreds of samples from both models, and displayed them in random order to a third set of users. Data was collected "in the wild", using personal webcams, without supervision, and users had no knowledge of the experiment. We obtained evaluation data from 76 users, spanning 536 sketches. Figure 2 shows the results of the evaluation, indicating that all of the facial expression metrics improved in the expected direction under the LC-GAN. Two of the metrics reached statistical significance: mean amusement, $t(535) = 2.31, p < .05$, and mean sadness, $t(535) = -2.01, p < .05$.

In conclusion, we have demonstrated that implicit social feedback in the form of facial expressions not only can reflect user preference, but also can significantly improve the performance of a deep learning model when it is evaluated on a much larger, independent set of users.

ACKNOWLEDGMENTS

We would like to thank James Tolentino, Ira Blossom, Adrien Baranes, Katherine Lee, Chris Han, Curtis Hawthorne, Rebecca Salois, Josh Lovejoy, Sherol Chen, and Mike Dory for their contributions to this project.

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
