# OpenReview forum: "Learning via social awareness: improving sketch representations with facial feedback"
_ICLR.cc/2018/Workshop — Accept_

### Official Review · AnonReviewer3 · 2018-03-09
**This submission proposes to utilize social interaction, facial expressions in this submission, to improve deep learning model output. It is an interesting topic, although it lacks some important technical details.**

**Rating:** 6
**Confidence:** 5

**Review:**

This submission proposes to utilize social interatction, facial expressions in this submission, to improve deep learning model output.

-- Pros:
   -- This topic is very intestesting. It tries to utilze the human interatctions to improve AI agent learning.

-- Cons:
   -- Some technical details are missing in this submission. For example, the configuration of LC-GAN and Sketch RNN used in this paper are not mentioned. So it might be a little early to achieve a conclusion from Figure 2.

   -- The author doesn't provide details about what facial expression detector they are using in this work,  and how accurate the detector is.

---

### Official Review · AnonReviewer1 · 2018-03-11
**interesting research direction, expected results**

**Rating:** 6
**Confidence:** 4

**Review:**

In this work the authors utilize a latent constraints GAN to produce embeddings for the variational Sketch-RNN model, that are likely to produce drawings leading to positive facial expressions.  The authors train the LC-GAN to produce embeddings that when utilized with sketch-RNN, produce sketches that maximize positive and minimize negative emotions.

On the negative side of things, there not enough details on the facial expression recognition system utilized, although the authors show that the results correlate well with the self-reported emotions.   The evaluation part of the paper could be further improved (e.g., discuss the facial expression metrics that are mentioned).

Due to the methods used, I feel that the results are expected (i.e., this is what LC-GAN is supposed  to do - map embeddings maximizing a value function).  Nevertheless, I think that it is a very interesting research direction: to incorporate implicit human social feedback in AI.

---

### Decision · Program_Chairs · 2018-03-20
**ICLR 2018 Workshop Acceptance Decision**

**Decision:**

Accept

**Comment:**

Congratulations, your paper was accepted to the ICLR workshop.